# Total Fatty Acid and Polar Lipid Species Composition of Human Milk

**DOI:** 10.3390/nu14010158

**Published:** 2021-12-29

**Authors:** Talat Bashir Ahmed, Merete Eggesbø, Rachel Criswell, Olaf Uhl, Hans Demmelmair, Berthold Koletzko

**Affiliations:** 1Division of Metabolic and Nutritional Medicine, Dr. von Hauner Children’s Hospital, LMU Klinikum, Ludwig-Maximilians-Universität, 80337 Munich, Germany; T.Ahmed@med.uni-muenchen.de (T.B.A.); Olaf_Uhl@web.de (O.U.); 2Department of Environmental Health, Norwegian Institute of Public Health, P.O. Box 222 Skøyen, 0213 Oslo, Norway; Merete.Eggesbo@fhi.no (M.E.); rachelcriswell@gmail.com (R.C.)

**Keywords:** human milk, fat content, fatty acid composition, phospholipids, phosphatidylcholine, sphingomyelin, arachidonic acid, docosahexaenoic acid

## Abstract

Human milk lipids are essential for infant health. However, little is known about the relationship between total milk fatty acid (FA) composition and polar lipid species composition. Therefore, we aimed to characterize the relationship between the FA and polar lipid species composition in human milk, with a focus on differences between milk with higher or lower milk fat content. From the Norwegian Human Milk Study (HUMIS, 2002–2009), a subset of 664 milk samples were analyzed for FA and polar lipid composition. Milk samples did not differ in major FA, phosphatidylcholine, or sphingomyelin species percentages between the highest and lowest quartiles of total FA concentration. However, milk in the highest FA quartile had a lower phospholipid-to-total-FA ratio and a lower sphingomyelin-to-phosphatidylcholine ratio than the lowest quartile. The only FAs associated with total phosphatidylcholine or sphingomyelin were behenic and tridecanoic acids, respectively. Milk FA and phosphatidylcholine and sphingomyelin species containing these FAs showed modest correlations. Associations of arachidonic and docosahexaenoic acids with percentages of phosphatidylcholine species carrying these FAs support the conclusion that the availability of these FAs limits the synthesis of phospholipid species containing them.

## 1. Introduction

Human milk is the optimal source of infant nutrition during the first 4–6 months of life, and provides nutrients, bioactive compounds, and immunological factors to support healthy growth and development [1,2,3,4].

Fat contributes around half of the energy intake of breastfed infants, and also provides indispensable polyunsaturated fatty acids (PUFAs) and lipid-soluble vitamins [2,5]; it is the most variable macronutrient in human milk, with content ranging from 2.4 to 5.5 g/100 mL (10th to 90th percentiles) [6]. In human milk, lipids are emulsified as milk fat globules (MFGs) that consist of a triacylglycerol (TAG)-rich core surrounded by a tri-layered milk fat globule membrane (MFGM) comprising glycerophospholipids, sphingolipids, cholesterol, and proteins [5,7,8]. Previously the importance of milk LCPUFAs—including n-6 arachidonic acid (ARA) and n-3 docosahexaenoic acid (DHA)—for infant development has been recognized [9], but only recently has the importance of the complex polar lipids in the MFGM for infants’ gut maturation and the establishment of the intestinal immune system been described [10]. Positive effects of MFGM supplementation on infants’ cognitive development and infection risks suggest that individual components of MFGM (e.g., choline, sphingomyelin, gangliosides, and cholesterol) may play important roles in the neurodevelopment and immune system maturation of infants [11]. Human milk has been recognized as a good source of sphingolipids, and it may well be that the contribution of dietary sphingolipids to infant development has been underestimated [12]. The glycerophospholipids primarily contain choline or ethanolamine and, to a lesser extent, inositol or serine as polar head groups [7,13]. A considerable variety of phospholipid (PL) species arises from the different fatty acids (FAs) at the sn-1 and sn-2 positions of the glycerol backbone, which can be bound via ester or ether bonds [14,15,16]. Sphingolipids consist of a sphingosine backbone with a FA bound to its amino group. The binding of phosphocholine forms sphingomyelin (SM)—a major polar lipid component in both cell membranes and MFGMs [17,18].

Milk TAG and glycerophospholipids are synthesized at the endoplasmic reticulum of mammary gland alveolar cells [19]; they combine to form MFGs of different sizes, which vary with fat content [15]. FAs are the main constituents of milk TAG and glycerophospholipids, and originate from maternal diet, fat stores, or de novo synthesis [20,21,22]. Roughly 98% of milk FAs are present in TAG; therefore, total milk FA composition reflects milk TAG composition, while only a small portion of FAs are esterified in PLs [23,24,25]. Diet has been identified as the main predictor of total milk FA composition, whereas maternal genotype and duration of lactation also influence PUFA and long-chain PUFA (LCPUFA) levels in particular [5]. It is an open question whether this also applies to the polar lipids of the MFGM, which becomes relevant with the now-known nutritional value of the polar lipids. Information on how important the determinants of total FA composition are for the polar lipids in human milk can be expected from studying the associations between FA and polar lipids. Wang et al. studied the total FA and PL-class-specific FA composition of human milk samples collected from 20 women during the first week of lactation. Most of the %FA in the PL subclasses tended to correlate positively with the corresponding percentages in total lipids, but statistical significance was found only for dihomo-gamma-linolenic acid (C20:3n-6) in phosphatidylethanolamine and alpha-linolenic acid (C18:3n-3) in phosphatidylserine [26]. A series of further influencing factors have been described [27], but to the best of our knowledge, it is still uncertain whether the human milk fat content is associated with milk FA composition. Limited availability of some components, such as LCPUFAs, could potentially affect total LCPUFA content and the contribution of specific polar lipids. If this is the case, higher human milk fat content would mean higher energy intake of the infant, but not necessarily higher intake of critical nutrients such as LCPUFAs or certain polar lipids.

Concerning bovine milk, Mesilati et al. reported FA compositional differences associated with different sizes of bovine MFGs, and suggested that differences in FA composition also affect MFGM polar lipid composition [28]. A negative correlation between the content of polar lipids and the diameter of MFGs has been reported in bovine MFGs [24,28,29].

In the present study, we aimed to investigate whether FA and polar lipid composition in human milk differs between milk samples with high and low-fat content, and whether milk total FA composition is associated with the milk polar lipid species phosphatidylcholine (PC) and SM.

## 2. Materials and Methods

This study analyzed human milk samples collected in the Norway-wide HUMIS cohort that enrolled 2606 mothers during 2002–2009. The cohort and the details of milk sampling have been described elsewhere [30,31,32]. Mothers were asked to collect and combine several 25 mL milk samples during the second month postpartum and then send the pooled milk samples to the Norwegian Institute of Public Health in Oslo, Norway, where they were stored at −20 °C. For the present study, we used a subset of samples obtained from mothers of singleton infants with available metabolomics and FA data (*n* = 789). Characteristics of the mothers are reported in Appendix A.

### 2.1. Measurements

Five milliliter aliquots of the milk samples were sent on dry ice to Hauner Children’s Hospital, Munich, Germany. Milk was thawed in a water bath at 20–30 °C, thoroughly mixed, and a 100 µL aliquot was mixed with 700 µL of methanol and 600 µL of methyl tert-butyl ether (containing triundecanoin as an internal standard) for total FA and polar lipid species analysis, vortexed for 30 s, and centrifuged at room temperature (1500× *g*) for 5 min to precipitate proteins. For the synthesis of FA methyl esters from free and lipid-bound FAs, 300 µL of the supernatant with dissolved lipids was transferred into a 4 mL glass vial and combined with methanol (1260 µL), hexane (140 µL), and methanolic hydrochloric acid (400 µL, 3M, Supelco, Deisenhofen, Germany). After mixing, the vial was heated to 100 °C for 1 h. Once cooled down to room temperature, phase separation was achieved via short centrifugation after the addition of 500 µL of water and 1000 µL of hexane. An aliquot of the upper, organic phase was used for gas chromatography with flame ionization detection after appropriate dilution (Agilent 7890 GC with Gerstel KAS 4 injector). A 50 m BPX-70 column (SGE, Weiterstadt, Germany) was used for gas chromatographic separation of FA methyl esters [33]. GLC569 (Nu-Chek-Prep Inc., Elysian, USA) was used for peak identification and quantification. FA data are presented as concentrations (g/L) and weight/weight percentages. In total, 32 FAs with a chain length of 8–22 carbon atoms were quantified. Eighty-two aliquots of a reference milk sample were randomly distributed among the study samples to check analytical performance. Averaged across all quantified FAs, the coefficient of variation (CV) was 13% for concentrations and 9% for percentages.

A further 100 µL aliquot of the supernatant was used for the analysis of acylcarnitines, lysophosphatidylcholine (lyso-PC), diacyl-phosphatidylcholines (PCaa), acyl-alkyl-phosphatidylcholines (PCae), and SM via flow-injection tandem mass spectrometry. Sample preparation was performed in 96-well plates. First, the aliquots were combined with 500 µL of methanol containing ammonium acetate (0.4 g/L) and, as internal standards, lyso-PC13:0, PCaa.C28:0, and a mixture of carnitine esters (C2, C8, and C16). After mixing at room temperature, 96-well plates were cooled to −20 °C for 20 min and centrifuged (2000× *g*) for 10 min. Then, 320 µL of the supernatants were transferred into an Agilent 96-well plate for injection (30 µL) into the flow-injection mass spectrometry system. Mass spectrometry was performed with a triple-quadrupole mass spectrometer (4000QTRAP, Sciex, Darmstadt, Germany) using an electrospray ionization source coupled with a liquid chromatography system (Agilent, Waldbronn, Germany). Analyst 1.5.1 software (Sciex, Darmstadt, Germany) was used for the measurements, which were performed as previously described for plasma samples [34,35]. Of the 330 molecular species quantified with this method, we included species if their average contribution exceeded 0.01 mol%. Thus, we considered 161 lipid species for further analysis, which were measured with highly variable precision. Together with the study samples, 66 aliquots of a milk sample were analyzed as quality controls. For higher concentrated species (24 PL species > 1 mol%), the mean coefficient of variation (CV) ranged between 9.3% and 13.1%. However, the CV was higher for lower concentrated species, and we excluded species that showed a CV > 100% in the control samples, leaving us with a total of 130 species with an average CV of 37%.

The applied flow-injection mass spectrometry technique does not indicate the position of double bonds or the distribution of carbon atoms between FAs. Thus, we applied the XX:Y nomenclature for PL species, where “XX” is the number of carbon atoms and “Y” is the combined number of double bonds in the FAs. In addition, if both FAs are bound via ester bonds, “aa” is used, while “ae” indicates the presence of an ether bond.

### 2.2. Data Treatment

Two analytical batches corresponding to two specific 96-well plates showed inconsistent results for quality control and study samples; therefore, we excluded these data (*n* = 125) from further analysis; finally, we included 664 samples. Outliers were identified via box–whisker plots, and any observation outside the range of Q1 − (1.5 IQR) and Q3 + (1.5 IQR) was defined as a potential outlier. Fifty observations in multiple metabolites were identified as outliers. We prepared an alternate dataset by excluding these 50 samples, yielding a dataset with 614 values. Correlation analyses were performed with both datasets to test for outliers’ influence on associations, which did not show any significant difference in correlation patterns between total FAs and PL subclasses and molecular PL species, respectively. Therefore, we used the complete dataset (*n* = 664) for the reported analyses.

The dataset was separated into quartiles according to total FA content, taken as a proxy for fat content, in order to study the influence of fat content on FAs and lipid species percentages. Concentrations and percentages are presented as means ± standard deviations.

### 2.3. Data Analysis

The associations between total milk FAs and choline-containing milk PL subclasses (lyso-PC, PC, and SM) were analyzed with Pearson’s correlation coefficients. Multiple linear regression analysis was used to study the role of total milk LCPUFAs as predicting factors for specific PL species, and findings are presented as β coefficients with *p*-values. Comparisons between high and low total fat quartiles were performed using Student’s *t*-test, and association analyses were partially repeated selectively for these quartiles.

All statistical tests were performed with SPSS Version 25.0; IBM Deutschland GmbH, (Ehningen, Germany). Statistical significance was accepted at *p* < 0.05. As a high number of tests were performed, *p*-values were adjusted for multiple comparisons using Bonferroni’s multiple testing correction (0.05/total number of tests). Significance levels were adjusted to a *p*-value of 0.0006 for all association analyses (total number of tests for correlations = 75, 0.05/75 = 0.0006), and to a *p*-value of 0.001 for *t*-tests (total number of *t*-tests = 29, 0.05/29 = 0.001). 

## 3. Results

The total FA composition of the studied milk samples was similar to previously reported data from European women; however, the mean DHA percentage content of 0.39% was higher than the previously report global mean of 0.32% [36], presumably reflecting a relatively high seafood and cod liver oil intake in the Norwegian population. In contrast, the mean ARA content of 0.36% was lower than the global mean of 0.47%.

Table 1 shows the percentage contribution of each FA to total FAs for the total sample (*n* = 664) and the lowest (14.6 ± 3.2 g/L) and highest (35.9 ± 6.1 g/L) total FA concentration quartiles. FA composition was similar in milk with low and high FA concentrations, with significant differences found only for 6 FAs, i.e., caprylic acid, (C8:0), tridecanoic acid (C13:0), pentadecanoic acid (15:0), behenic acid (C22:0), myristoleic acid (C14:1n-5), and palmitoleic acid (C16:1n-7).

The relative distribution of molecular species of PC and SM were calculated as percentages of total PC or SM. Molecular species with a contribution greater than 0.5% of total PC or SM are presented in Appendix A, respectively. We identified PCaa.C36:2 in PC and SM 40:1 in SM as the most abundant species. Only the percentages of 3 PC and 3 SM molecular species were significantly different between milk in the highest and lowest FA quartiles. In all cases, quantitative differences were small. 

We noted a gradually increasing trend in %PC of total PLs from Q1 to Q4, while %SM of total PLs showed a decreasing trend (Figure 1 and Figure 2). The comparison of the lowest and highest quartiles showed a higher ratio of total SM (µmol/L) to total FAs (g/L) (6.04 ± 1.64 vs. 4.13 ± 0.88 µmol SM/g FA; *p* = 1.167 × 10^−^³⁰) and of total PC (µmol/L) to total FAs (g/L) (2.80 ± 0.89 vs. 2.25 ± 0.18 µmol PC/g FA; *p* = 5.77 × 10^−^¹⁰) in the lowest total FA quartile, but a higher percentage contribution of SM to total PLs in the lowest FA quartile (68.56 ± 5.55% vs. 64.81 ± 4.98%; *p* = 3.77 × 10^−^¹⁰), corresponding to a significantly lower PC percentage (27.19 ± 5.06% vs. 30.32 ± 4.80%; *p <* 0.001, *p* = 3.76 × 10^−^¹⁰), in the lowest total FA quartile.

### 3.1. Association of Concentrations and Composition of Choline-Containing PL Subclasses and Total FAs

Associations between total FAs and choline-containing PL subclasses in milk are shown as concentrations and percentages in Table 2. Total FA concentration was significantly correlated with total lyso-PC, total PC, and total SM, with r-values of 0.51, 0.65, and 0.66, respectively. Moreover, individual FAs were all significantly positively correlated with concentration totals of lipid classes (Table 2). However, when individual FAs and PL subclasses were expressed as percentage values, only C22:0 and C13:0 percentages were positively correlated with %SM, and negatively correlated with %PC.

### 3.2. Associations of Total FAs with Choline-Containing PL Species

We calculated Pearson’s correlation coefficients for some plausible combinations of total FA percentages with individual PC and SM species, presumably containing the corresponding FAs (Table 3). FAs with the same number of carbon atoms and double bonds at different positions were combined for the correlation analysis, e.g., C18:1n-9 and C18:1n-7 to total C18:1. For PCs, we found significant positive associations between saturated FAs and LCPUFAs in PC species containing the corresponding saturated FAs and LCPUFAs. Interestingly, no significant associations were detected for PC species assumed to contain essential FAs and the corresponding milk total essential FA percentages.

For the associations of FAs with SM species (Table 4), we assumed that in each SM species 18 carbon atoms and 1 double bond were contributed by the sphingosine backbone. Low-to-moderate-significance positive associations were observed between SM species and corresponding saturated FAs and the monounsaturated FA C18:1. There were no significant associations between SM species and corresponding polyunsaturated FAs.

For PC species assumed to contain ARA, eicosapentaenoic acid, or DHA, we performed a more detailed investigation considering these FAs as predictors of individual PC species, using multiple linear regression. The respective LCPUFAs were significant predictors for all of these PC species, while percentages of the saturated FAs were not significantly associated with the PC percentage (Table 5). The regression analyses were repeated separately for the highest and lowest total fat quartiles, which showed similar results (Appendix A).

## 4. Discussion

The observed percentages of total milk FAs were consistent with ranges reported for European populations, with a trend towards higher n-3 LCPUFA, which appears to reflect a relatively high intake of fish and cod liver oil in Norway [37]. For all major FAs, the total FA content was not associated with the relative FA composition, nor with the molecular species composition of PC and SM.

In our study, the relative contribution of individual FAs showed statistically significant differences between the highest and the lowest quartiles for only 6 FAs. All of these were minor components that are not expected to influence the nutritional value of the milk appreciably. The contribution of palmitoleic acid (C16:1n-7) was 2.4% in the lowest FA quartile compared to 2.7% in the highest quartile, which could reflect a higher stearoyl-CoA desaturase-1 enzyme activity when FA availability is high [34,38]. This could also explain the higher myristoleic acid (C14:1n-5) contribution in the highest FA quartile, but differences were marginal (0.26% vs. 0.24%). Among the saturated FAs, there were slightly lower percentages of the odd-chain FAs C13:0 and C15:0 in the highest FA quartile, which may indicate that these FAs are not produced by mammalian metabolism, but are derived from the maternal dietary intake of ruminant milk or meat, along with dietary fiber metabolized by maternal gut bacteria [39,40]. Consequently, increased incorporation of other FAs into milk fat will lead to a reduced percentage of these FAs, with limited availability. Percentages of all medium-chain FAs tended to be higher in the lowest FA quartile, but only for C8:0 was significance observed (0.4 vs. 0.3%). Thus, increased requirements for milk with a higher FA content may more easily be met with the uptake of preformed FAs than with increased mammary gland synthesis of medium-chain FAs. We could not find an obvious explanation for the higher C22:0 percentage in the lowest total FA quartile, but it may be related to the relatively higher SM content in the low-FA milk and the positive association between SM and C22:0 %. There were no significant differences for the major milk FAs palmitic acid (C16:0) and oleic acid (C18:1n-9) and all PUFAs, including LCPUFAs, although a tendency towards higher n-3 LCPUFA % in the lowest FA quartile was found. This could be related to the higher PL-to-total-FA ratio in the lowest FA quartile, and the fact that LCPUFA percentages tend to be higher in phosphatidylethanolamine, phosphatidylinositol, and phosphatidylserine than in total milk fat; typically, in PC, the LCPUFA % is higher than in total milk FA by a factor of two or three [26,41].

In agreement with previous studies, we found milk SM to be more abundant than PC [12,13,15]. The comparison of the quartiles regarding the ratio of PC or SM to total FAs, showed higher ratios in the lowest FA quartile. This is consistent with the observations in bovine milk, and supports the concept that higher fat content is associated with larger MFGs, which have less surface area and fewer MFGM phospholipids relative to their mass [42,43]. In the study by Selvalatchmanan et al., individual lipid species were measured via liquid chromatography–mass spectrometry, and the comparison of milk samples collected from the same mothers in the morning and the evening revealed higher concentrations in the evening for most of the species. The smaller increase in the polar lipid species compared to TAG species from morning to evening milk confirms our finding of a lower polar-lipid-to-TAG ratio in the highest FA quartile [43]. Interestingly, this study also observed that the morning–evening increase was highest for phosphatidylethanolamine, intermediate for PC, and smallest for SM within the polar lipids. This agrees with the observation in our study that the relative contribution of SM to total polar lipids was lower in the highest FA quartile. The biosynthesis of TAG and PC both take place at the endoplasmic reticulum, and start from the same precursors glycerophosphate and FAs (Figure 3) [44]. This is compatible with a good correlation between total FA and PL content, and the under-proportional increase in PLs could be explained by limited enzymatic capacity in the Kennedy pathway, limited availability of choline, or extensive conversion of PC to other PL classes. A further plausible assumption could be that in PC, the contribution of short- and medium-chain FAs is much lower than in TAG, and as a consequence, the availability of preferred substrates could become limiting for PC synthesis, but not for TAG, leading to a lower PC/TAG ratio in higher FA content milk.

With higher FA content, the PC/SM ratio is shifted towards PC, suggesting that membrane properties are different between high and low-FA-content milk, with a trend towards less and less rigid lipid rafts due to a relatively lower content of SM and potentially also cholesterol of larger MFGs in high-fat milk. However, the difference was statistically significant but small, so it is uncertain whether this influences the digestion of milk fat in infants.

For the relative distribution of choline-containing molecular species, this study identified PCaa.C36:2 as the most abundant PC species, followed by PCaa.C32:0, PCaa.C34:0, PCaa.C34:1, PCaa.C34:2, and PCaa.C36:1. For SM, we identified SM.C40:1 as the most abundant species, followed by SM.C34:1, SM.C36:1, SM.C38:1, SM.C42:1, and SM.C42:2. These findings can be compared to PC and SM species composition data reported for milk samples collected in Singapore, China, and Germany [10,45,46]. Although proportions were different, the most frequent species were the same as in our samples collected in Norway. This indicates that differences in dietary habits, which lead to differences in the milk FA composition [27], are largely compensated by the specific substrate preferences of the SM and PC synthesis pathways.

In agreement with the similarity of the FA composition of the highest and lowest FA quartiles, differences in the PC and SM molecular species composition were only marginal. This fits with the observations in cell culture studies with permeabilized rat hepatocytes, which indicated that diacylglycerol acyltransferase and choline phosphotransferase share a common precursor pool of diacylglycerols when producing TAG or PC, respectively, leading to the conclusion that the availability of diacylglycerol may limit the synthesis of both PC and TAG [47].

A further objective of this study was the description of the association between the FA composition of TAG and (a) the percentage contribution of PL subclasses SM, PC, and lyso-PC to total PLs, along with (b) the contribution of individual molecular species to total PLs. In respect to objective (a) we investigated associations based on absolute concentrations and relative compositional data. When data were expressed as concentrations, total FAs were correlated with PC and SM, with r-values above 0.6. This was expected because with increasing fat content, more emulsifier is required, even if the requirement is reduced by the larger diameter of the MFGs with increasing fat content. The correlations are also significant for all individual FA concentrations, with r-values above 0.4 except for C22:5n-6, C20:5n-3, C22:5n-3, C22:6n-3, and C13:0 (with PC only), which showed lower correlations. However, these FAs each contribute less than 0.5% to total FAs, so they hardly affect total fat content and, thus, their correlation with polar lipids can be small. On the other hand, n-3 LCPUFA was shown to be highly dependent on dietary intake [27,48,49,50,51], which could lead to lower correlations. The associations of FA concentrations with PC and SM were relatively similar, and tended to be lower with lyso-PC. This is compatible with the assumption that lyso-PC species are derived from PC, with some additional variability introduced by hydrolysis activity.

Using FA percentages, there were no significant associations with PLs except for C22:0 and C13:0, which were both correlated positively with % total SM and negatively with %PC. However, the r-values were only around 0.2 and −0.2, respectively; thus, the SM-to-PC ratio did not appreciably change with the milk FA composition. Furthermore, SM.C40:1 was the most abundant SM species, indicating a preferential use of C22:0 as a substrate by dihydroceramide synthase; thus, the higher availability of C22:0 could increase the overall synthesis of dihydroceramide from sphingosine and saturated FAs to some degree. We did not identify an apparent reason for the positive association of C13:0 with SM or the negative association with PC, but it might have been due to the positive association between C22:0 and C13:0 percentages in total FAs. Finally, it is worth mentioning that there was no association between C16:0 and SM, which indicates that the availability of palmitic acid does not limit sphingosine synthesis, or that sphingosine does not limit SM synthesis. FA composition was not associated with the level of total SM, in agreement with an observation by Dei Cas et al. that the analyzed milk sphingolipid species including SM, ceramide, and dihydroceramide [12]. They were not different between mothers who consumed a Mediterranean, a vegetarian, or a carnivorous diet, according to a multivariate analysis. [12]. Although not reported, it can be assumed that diet induces differences in total milk FAs, but maternal metabolism establishes a sphingolipid supply for the infant independent of the diet. This suggests the importance of sphingolipid intake of the infant and testing of infant formulas with improved sphingolipid content [12].

Although FA composition is not related to the SM/PC ratio, there are some associations between the FA percentages and corresponding SM or PC species percentages, which include these FAs. The correlation analyses showed significant positive associations for most FAs, with a maximal r-value of 0.54 between eicosapentaenoic acid (C20:5) and PCaa.C36:5. However, r-values were generally around 0.2 and, interestingly, there seemed to be no significant associations between essential FAs and PC species, which presumably contained these FAs. FAs are taken up from the circulation, contributing to TAG and polar lipids, and as synthesis of TAG, PC, and SM takes place at least partially at the endoplasmic reticulum, close correlations could be expected. On the other hand, the contribution of medium-chain FA synthesis and incorporation increases with the availability of glucose, or decreases with the availability of long-chain FAs for uptake from the circulation. As there is a strong preference for the incorporation of medium-chain FAs into TAG, this variability only affects TAG composition, and attenuates the association between lipid classes.

The most plausible reason for the absence of significant associations of C18:3 percentages with the identified PC species containing this FA seems to be that we could not unambiguously identify the PC species with three double bonds; PCaa.C34:3 might be a mixture of species combining C16:0/C18:3, C14:0/C20:3, and C16:1/C18:2. For PCaa.C36:3, the FA combinations C16:0/C20:3 and C18:1/C18:2 seem as plausible as C18:0/C18:3. In both cases, too much contribution of other FAs could obscure the influence of C18:3 percentage, which is only ~1% of the total FAs. Although C18:2n-6 is with ~12% more plentiful here, C18:1—the most abundant FA, at 34%—contributes C18:1/C18:1 and C16:1/C18:1, respectively, and might obscure associations of C18:2n-6 with PCaa.C36.2 and PCaa.C34.2.

PC species contain two FAs, and it can be asked whether both FAs are equally associated with the percentages of the PC species. As LCPUFA supply is of specific importance for infant development, and as there might be differences in the absorption and metabolic disposition between TAG and polar-lipid-bound LCPUFAs [52,53], we focused on associations between LCPUFAs and corresponding PC species (Table 5). It turned out that only the LCPUFA percentage was highly significantly associated with the PC species in all cases. Compared to the importance of the LCPUFAs, the influences of the saturated FAs seemed marginal, as the standardized betas for the saturated FA were close to 0. The saturated FAs C16:0 and C18:0 together contribute more than 30% to total FAs, and occur in combination with the LCPUFAs, but in much higher abundance together with other FAs. On the other hand, the LCPUFAs ARA, eicosapentaenoic acid, and DHA together contribute less than 1% to total FAs, and the combinations with C16:0 and C18:0 are the most abundant species with LCPUFAs. Thus, the LCPUFAs limit the contribution and define the percentage contribution to PLs. Nevertheless, the associations were not very close, as for none of the PC species could both FAs together explain more than 30% of the variance—and in most cases, it was around 10%. In agreement with the observation that FA percentages and the species composition of SM and PC seemed not to be influenced by the fat content, the regression coefficients did not differ between the high- and low-fat quartiles.

### Strength and Limitations

This is the first study investigating the associations of large numbers of individual choline-containing PL species with total human milk FAs and the effect of fat content on FA and PL molecular species composition in a sizeable cohort. This enabled the identification of minor differences and associations. A limitation of the applied analytical technique was that molecular species could not unambiguously be identified as, in some cases, multiple species probably contributed to the mass transition used for identification. In addition, as the analyses were limited to choline-containing species, phosphatidylethanolamine molecular species—which are major components of MFGMs could not be quantified, and thus the reported percentages are not fully representative. Milk storage at −20 degrees likely led to alterations during storage, including hydrolysis of lipids and (per) oxidation of FAs. Studies with human donor milk suggest that oxidation of long-chain polyunsaturated fatty acids is limited, and FA composition should not be significantly changed during sample storage [54]. On the other hand, hydrolysis of lipids including PL during storage and thawing could have been extensive [55]. As glycerophospholipids seem less stable than SM, the observed SM percentage could be an overestimation. Nevertheless, our findings with respect to the associations between FA and lipid species and the marginal influence of fat content should be valid.

## 5. Conclusions

With increasing total FA content in human milk, the ratio of PL to total milk FAs decreases, which appears to reflect larger MFGs with a lower PL-to-TAG ratio. While total FA content showed only minor relationships with FA and PC and SM molecular species composition, the contribution of SM relative to PC to milk PL decreased with increasing FA content. This could indicate that factors other than the availability of FAs, such as enzymatic synthesis or intracellular transport capacity, may limit SM synthesis. FA composition only marginally influences the contribution of PC and SM to total PL. Nevertheless, there are modest but significant positive associations between total milk FAs and corresponding PC and SM species. Human milk FA composition is strongly influenced by maternal diet, as far as triglyceride FAs are concerned, but according to our study, this does not necessarily apply for FAs incorporated into PC or SM. Further studies to identify genetic or metabolic factors influencing milk polar lipids along with studies related to the association between milk fat content and composition, considering potential effects on digestion and infant development—seem to be warranted.

## Figures and Tables

**Figure 1 nutrients-14-00158-f001:**
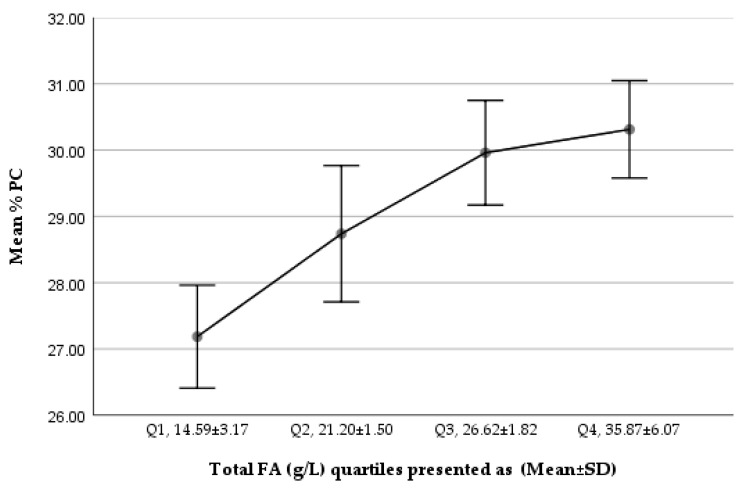
Trend of %PC of total PLs in Q1–Q4 (total FA g/L) of the studied milk samples.

**Figure 2 nutrients-14-00158-f002:**
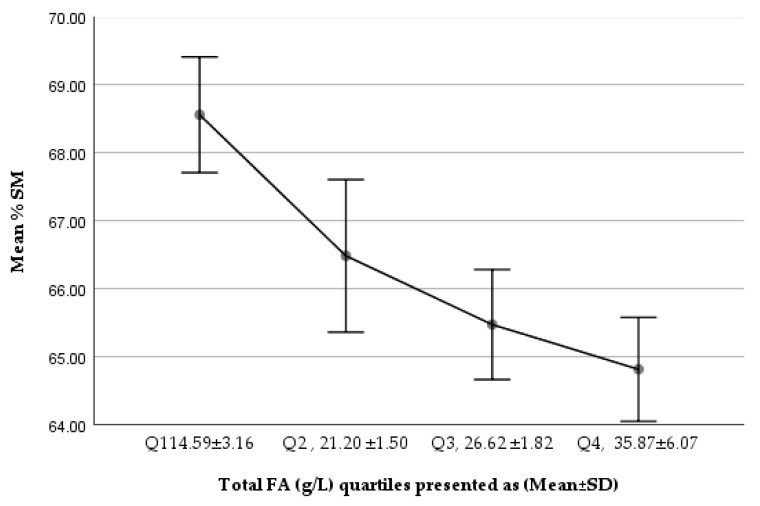
Trend of %SM of total PLs in Q1–Q4 (total FA g/L) of the studied milk samples.

**Figure 3 nutrients-14-00158-f003:**
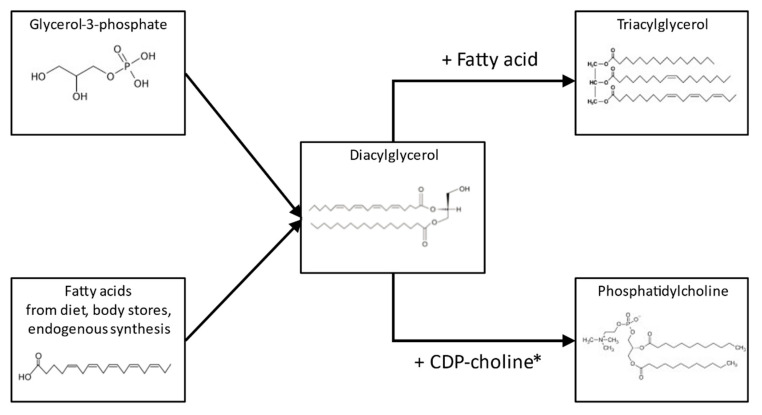
Simplified scheme (including examples of molecular structures) of the synthesis of triacylglycerols and phosphatidylcholines from fatty acid and glycerol-3-phosphate precursors, emphasizing the potential common intermediate diacylglycerol. *: Cytidine diphosphate choline.

**Table 1 nutrients-14-00158-t001:** Fatty acid percentages (mean ± SD) of the total studied HUMIS milk samples and samples in the lowest and highest quartiles of total FAs; *p*-values are derived from *t*-tests comparing the quartiles.

FA	Total Sample (*n* = 664)	Lowest FA Quartile (*n* = 166)	Highest FA Quartile (*n* = 166)	*p*-Value
Total fatty acid (FA) content (g/L)	24.57 ± 8.60	14.59 ± 3.17	35.87 ± 6.08	**2.85 × 10^−110^**
**Saturated FAs** Caprylic acid (C8:0)	0.34 ± 0.08	0.40 ± 0.10	0.29 ± 0.06	**1.20 × 10^−23^**
Capric acid (C10:0)	1.63 ± 0.35	1.68 ± 0.37	1.58 ± 0.30	0.008
Lauric acid (C12:0)	5.93 ± 1.65	5.99 ± 1.71	5.80 ± 1.53	0.273
Tridecanoic acid (C13:0)	0.11 ± 0.03	0.15 ± 0.05	0.08 ± 0.02	**8.34 × 10^−39^**
Myristic acid (C14:0)	6.00 ± 1.39	6.07 ± 1.59	5.92 ± 1.30	0.355
Pentadecanoic acid (C15:0)	0.36 ± 0.09	0.38 ± 0.11	0.34 ± 0.08	**0.001**
Palmitic acid (C16:0)	23.00 ± 1.64	22.91 ± 1.80	23.17 ± 1.50	0.155
Margaric acid (C17:0)	0.31 ± 0.04	0.30 ± 0.04	0.31 ± 0.05	0.022
Stearic acid (C18:0)	9.09 ± 1.30	9.22 ± 1.22	8.94 ± 1.16	0.029
Behenic acid (C22:0)	0.15 ± 0.04	0.16 ± 0.05	0.13 ± 0.02	**2.11 × 10^−9^**
**Monounsaturated FAs**				
Myristoleic acid (C14:1n-5)	0.25 ± 0.06	0.24 ± 0.07	0.26 ± 0.05	**2.71 × 10^−4^**
Pentadecenoic acid (C15:1n-5)	0.08 ± 0.02	0.08 ± 0.02	0.08 ± 0.02	0.850
Palmitoleic acid (C16:1n-7)	2.48 ± 0.64	2.40 ± 0.61	2.66 ± 0.64	**1.37 × 10^−4^**
Vaccenic acid (C18:1n-7)	1.65 ± 0.22	1.66 ± 0.23	1.65 ± 0.22	0.510
Oleic acid (C18:1n-9)	31.85 ± 2.50	31.52 ± 2.45	32.28 ± 2.39	0.005
Eicosenoic acid (C20:1n-9)	0.46 ± 0.09	0.48 ± 0.12	0.44 ± 0.08	0.004
**Polyunsaturated FAs**				
Mead acid (C20:3n-9)	0.02 ± 0.01	0.02 ± 0.01	0.02 ± 0.01	0.084
**n-6PUFA**				
Linoleic acid (C18:2n-6)	12.48 ± 2.55	12.47 ± 2.61	12.27 ± 2.63	0.473
Gamma-Linolenic acid (C18:3n-6)	0.13 ± 0.03	0.12 ± 0.03	0.13 ± 0.04	0.024
Eicosadienoic acid (C20:2n-6)	0.49 ± 0.15	0.49 ± 0.17	0.49 ± 0.14	0.964
Dihomo-gamma-linolenic acid (C20:3n-6)	0.35 ± 0.08	0.34 ± 0.08	0.36 ± 0.08	0.180
Arachidonic acid (C20:4n-6)	0.36 ± 0.07	0.36 ± 0.07	0.37 ± 0.07	0.432
Adrenic acid (C22:4n-6)	0.07 ± 0.03	0.07 ± 0.04	0.07 ± 0.018	0.527
Osbond acid (C22:5n-6)	0.02 ± 0.01	0.03 ± 0.01	0.02 ± 0.01	0.054
**n-3PUFA**				
Alpha-Linolenic acid (C18:3n-3)	1.07 ± 0.29	1.07 ± 0.31	1.03 ± 0.29	0.194
Eicosatrienoic acid (C20:3n-3)	0.05 ± 0.01	0.05 ± 0.01	0.05 ± 0.01	0.050
Eicosapentaenoic acid (C20:5n-3)	0.10 ± 0.07	0.11 ± 0.08	0.09 ± 0.07	0.030
Clupanodonic acid (C22:5n-3)	0.17 ± 0.06	0.17 ± 0.06	0.16 ± 0.05	0.061
Docosahexaenoic acid (C22:6n-3)	0.39 ± 0.21	0.42 ± 0.24	0.36 ± 0.19	0.013

Bonferroni adjusted *p*-value = 0.001. Data presented as mean ± SD. Fatty acid values are expressed as wt%.

**Table 2 nutrients-14-00158-t002:** Pearson’s correlation coefficients of total FA concentrations (g/L) and percentages with subclasses of PL concentrations (µmol/L) and percentages.

	Concentrations			Percentages		
FAs	Lyso-PC	PC	SM	%Lyso-PC	%PC	%SM
Caprylic acid (C8:0)	0.473 *	0.585 *	0.605 *	−0.026	−0.089	0.090
Capric acid (C10:0)	0.506 *	0.623 *	0.646 *	0.041	0.039	−0.047
Lauric acid (C12:0)	0.464 *	0.551 *	0.614 *	0.020	−0.040	0.032
Tridecanoic acid (C13:0)	0.259 *	0.383 *	0.425 *	−0.170 *	−0.219 *	0.248 *
Myristic acid (C14:0)	0.495 *	0.591 *	0.640 *	0.030	−0.038	0.028
Pentadecanoic acid (C15:0)	0.386 *	0.524 *	0.517 *	−0.051	0.023	0.035
Palmitic acid (C16:0)	0.491 *	0.632 *	0.638 *	−0.005	0.076	−0.070
Margaric acid (C17:0)	0.451 *	0.588 *	0.595 *	−0.001	0.006	−0.005
Stearic acid (C18:0)	0.460 *	0.564 *	0.599 *	−0.009	−0.121	0.115
Behenic acid (C22:0)	0.423 *	0.471 *	0.549 *	−0.035	−0.206 *	0.201 *
Oleic acid (C18:1n-9)	0.485 *	0.615 *	0.620 *	0.063	0.072	−0.084
Linoleic acid (C18:2n-6)	0.383 *	0.501 *	0.530 *	−0.085	−0.052	0.070
Alpha-Linolenic acid (C18:3n-3)	0.350 *	0.432 *	0.463 *	−0.009	−0.063	0.061
Mead acid (C20:3n-9)	0.386 *	0.520 *	0.517 *	−0.024	0.022	−0.014
Dihomo-gamma-linolenic acid (C20:3n-6)	0.445 *	0.591 *	0.616 *	−0.028	0.051	−0.041
Arachidonic acid (C20:4n-6)	0.444 *	0.611 *	0.596 *	−0.051	0.099	−0.079
Adrenic acid (C22:4n-6)	0.375 *	0.507 *	0.533 *	−0.057	−0.026	−0.039
Osbond acid (C22:5n-6)	0.201 *	0.373 *	0.325 *	−0.123	0.001	0.031
Eicosatrienoic acid (C20:3n-3)	0.433 *	0.524 *	0.559 *	0.014	−0.056	0.048
Eicosapentaenoic acid (C20:5n-3)	0.174 *	0.282 *	0.197 *	−0.023	0.040	−0.031
Clupanodonic acid (C22:5n-3)	0.344 *	0.465 *	0.410 *	0.007	0.036	−0.035
Docosahexaenoic acid (C22:6n-3)	0.262 *	0.369 *	0.280 *	0.019	0.063	−0.064
Total FA	0.509 *	0.645 *	0.663 *	----	----	----

* Significant with Bonferroni adjusted *p*-value ≤ 0.0006. Lyso-PC: lysophosphatidylcholine; PC: phosphatidylcholine; SM: sphingomyelin.

**Table 3 nutrients-14-00158-t003:** Pearson’s correlation coefficients of %FAs and %PCs of total PLs (*n* = 664).

%FAs	%PCs	r
**Saturated FAs**		
C16:0	PCaa.C32:0 (16:0/16:0)	0.272 *
C18:0	PCaa.C36:0 (18:0/18:0)	0.170 *
**Monounsaturated FAs**		
C16:1n-7	PCaa.C32:1 (16:0/16:1)	0.429 *
C18:1	PCaa.C34:1 (16:0/18:1)	0.243 *
C20:1n-9	PCaa.C40:1 (20:0/20:1)	0.268 *
**Essential FAs**		
C18:2n-6	PCaa.C34:2 (16:0/18:2)	0.032
C18:2n-6	PCaa.C36:2 (18:0/18:2)	0.092
C18:3	PCaa.C34:3 (16:0/18:3)	0.097
C18:3	PCaa.C36:3 (18:0/18:3)	−0.045
**Long-chain polyunsaturated FAs (LCPUFAs)**		
C20:4n-6	PCaa.C38:4 (18:0/20:4)	0.411 *
C20:4n-6	PCaa.C40:4 (20:0/20:4)	0.138 *
C20:5n-3	PCaa.C36:5 (16:0/20:5)	0.541 *
C20:5n-3	PCaa.C38:5 (18:0/20:5)	0.301 *
C22:6n-3	PCaa.C38:6 (16:0/22:6)	0.248 *
C22:6n-3	PCaa.C40:6 (18:0/22:6)	0.379 *

* Significant with Bonferroni adjusted *p*-value ≤ 0.0006. FAs: fatty acids; PLs: phospholipids; PCs: phosphatidylcholine species.

**Table 4 nutrients-14-00158-t004:** Pearson’s correlation coefficients of %FAs and %SM of total PLs (*n* = 664).

%FAs	%SM	r
**Saturated FAs**		
C15:0	SM.C33:1	0.202 *
C16:0	SM.C34:1	0.108
C17:0	SM.C35:1	0.435 *
C18:0	SM.C36:1	0.226 *
C22:0	SM.C40:1	0.208 *
**Unsaturated FAs**		
C16:1n-7	SM.C34:2	0.115
C18:1	SM.C36:2	0.223 *
C18:2n-6	SM.C36:3	−0.149
C20:1n-9	SM.C38:2	−0.058
C20:2n-6	SM.C38:3	−0.084
C22:5	SM.C40:6	−0.011

* Significant with Bonferroni adjusted *p*-value ≤ 0.0006.

**Table 5 nutrients-14-00158-t005:** Multiple linear regression analysis relating LCPUFA-containing PC species percentages to total FA percentages.

Dependent Variable (%PCs)	Independent Variable (%FAs)	R²	B (95%CI)	β	*p*-Value
%PCaa.C36:4		0.018			
	%C16:0		0.003 (−0.004–0.009)	0.034	0.381
	%C20:4n-6		0.259 (0.111–0.406)	**0.133**	**0.001**

%PCaa.C38:4		0.174			
	%C18:0		−0.008 (−0.016–0.000)	−0.074	0.041
	%C20:4n-6		0.785 (0.643–0.926)	**0.395**	**1.63 × 10^−25^**

%PCaa.C36:5		0.301			
	%C16:0		0.001 (0.000–0.001)	0.092	0.005
	%C20:5n-3		0.091 (0.080–0.101)	**0.543**	**1.53 × 10^−5^**

%PCaa.C38:5		0.095			
	%C18:0		−0.004 (−0.008–0.000)	−0.006	0.007
	%C20:5n-3		0.299 (0.225–0.373)	**0.295**	**9.29 × 10^−15^**

%PCaa.C38:6		0.063			
	%C16:0		0.001 (−0.001–0.003)	0.039	0.298
	%C22:6n-3		0.055 (0.039–0.072)	**0.250**	**7.02 × 10^−11^**

%PCaa.C40:6		0.148			
	%C18:0		−0.003 (−0.007–0.000)	−0.068	0.062
	%C22:6n-3		0.112 (0.091–0.134)	**0.370**	**8.48 × 10^−23^**

%PCaa: %diacyl-phosphatidylcholine; %FAs: % fatty acids; R²: coefficient of determination; B(%CI): unstandardized coefficient (95% confidence interval); β: standardized regression coefficient.

## Data Availability

Data cannot be made publicly available as the dataset contains sensitive and identifying information. The authors confirm that the data will be kept safe for 5 years, and will be made available upon request. Requests may be sent to the coordinator of Early Nutrition: B. Koletzko, von Hauner Children’s Hospital, Univ. of Munich Medical Center, Lindwurmstrasse 4, D-80337 Muenchen, Germany. Phone number 49-8944005-2826, email address Berthold.Koletzko@med.uni-muenchen.de or hans.demmelmair@med.uni-muenchen.de.

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
