# Peer review of "Total Fatty Acid and Polar Lipid Species Composition of Human Milk"

_nutrients, 2021, doi:10.3390/nu14010158_

Round 1

Reviewer 1 Report

This manuscript evaluated the total milk fatty acid (FA) composition, with particular attention to polar lipid species, in higher and lower fat samples from the Norwegian Human Milk Study. The authors found, among higher fat samples, that the phospholipid to total FA ratio and the sphingomyelin to phosphatidylcholine ratio were lower compared with the lowest FA quartile, likely related to the size of the milk fat globule and endogenous synthesis or transport capabilities, respectively. While this study is interesting, the authors provide little physiological or clinical context as to how these findings should be interpreted in the setting of infant health.

Minor:

  1. In the intro, please add a short explanation about why the fat content of human milk is important (beyond its role as pure energy, which the authors mention). In other words, if the authors find differences in the lipid species percentages, why should a reader interested in human milk, but not a part of the lipid field, find this interesting and important? Are there potential physiological effects on the infant if these percentages differ?
  1. The supplementary data was either not provided or reviewers did not have access to this. Would help to see that information.
  1. Have authors done any studies confirming polar lipid species, in particular, are not damaged beyond measurement with the freeze/thaw cycling? If there is some damage (would assume there is), how can this be accounted for in the percentages? Saw the line in 4.1 re this as a limitation, but can authors explain why only absolute concentration would be affected? What about differences in saturation?
  1. Would combine Table 1 subparts into a single table, even if it flows onto another page. Editors should be able to accommodate this.
  1. Find the results confusing when authors meander between FA in g/L to uM/g.
  1. Table 2—missing the * in caption explaining that the symbol represents significance.

Reviewer 2 Report

In this study, the authors investigated the composition of polar lipid species and total fatty acids in human milk. The sample size is good, and collection time crosses the different periods. Overall, the study design and performance are well. Some minor changes are necessary to make it better.

A change of milk fatty acid and polar lipid plays what kind of important role in the growth of infants or even animal studies should be briefly mentioned in the Introduction.

The number and unit should be separated by a space, such as 700μl vs. 600 μl in line 84.

All the abbreviations such as ARA should be listed for the first time in the manuscript.

Octanoic acid in line 166, caprylic acid in Table 1, even though they are the same thing.

Fig 1, Fig 2, and Fig 3 in the context are better changed to Figure 1, Figure 2, and Figure 3.

Supplementary data should be submitted together with manuscript.
